# Cytokine Profiling and Intra-Articular Injection of Autologous Platelet-Rich Plasma in Knee Osteoarthritis

**DOI:** 10.3390/ijms23020890

**Published:** 2022-01-14

**Authors:** Kanyakorn Riewruja, Suphattra Phakham, Patlapa Sompolpong, Rangsima Reantragoon, Aree Tanavalee, Srihatach Ngarmukos, Wanvisa Udomsinprasert, Tanyawan Suantawee, Sinsuda Dechsupa, Sittisak Honsawek

**Affiliations:** 1Osteoarthritis and Musculoskeleton Research Unit, Faculty of Medicine, Chulalongkorn University, King Chulalongkorn Memorial Hospital, Thai Red Cross Society, Bangkok 10330, Thailand; s.ung_ang@hotmail.com (K.R.); bebestthu@hotmail.com (S.P.); patlapas@mit.edu (P.S.); sinsuda.dech@gmail.com (S.D.); 2Program in Medical Science, Faculty of Medicine, Chulalongkorn University, King Chulalongkorn Memorial Hospital, Thai Red Cross Society, Bangkok 10330, Thailand; 3Immunology Division, Center of Excellence in Immunology and Immune-Mediated Diseases, Department of Microbiology, Faculty of Medicine, Chulalongkorn University, Bangkok 10330, Thailand; Rangsima.R@chula.ac.th; 4Vinai Parkpian Orthopaedic Research Center, Department of Orthopaedics, Faculty of Medicine, Chulalongkorn University, King Chulalongkorn Memorial Hospital, Thai Red Cross Society, Bangkok 10330, Thailand; areetang@orthochula.com (A.T.); srihatach@hotmail.com (S.N.); 5Department of Biochemistry, Faculty of Pharmacy, Mahidol University, Bangkok 10400, Thailand; wanvisa.udo@mahidol.ac.th; 6Department of Nutrition and Dietetics, Faculty of Allied Health Sciences, Chulalongkorn University, Bangkok 10330, Thailand; tanyawan.s@chula.ac.th

**Keywords:** chondrocytes, cytokines, knee, platelet-rich plasma, osteoarthritis

## Abstract

Osteoarthritis (OA) is a degenerative joint disease leading to joint pain and stiffness. Due to lack of effective treatments, physical and psychological disabilities caused by OA have a detrimental impact on the patient’s quality of life. Emerging evidence suggests that intra-articular injection of platelet-rich plasma (PRP) may provide favorable results since PRP comprises not only a high level of platelets but also a huge amount of cytokines, chemokines, and growth factors. However, the precise mechanism and standardization method remain uncertain. This study aimed to examine cytokine profiling in both PRP and platelet-poor plasma (PPP) of knee OA patients and to determine the effects of PRP on OA chondrocytes and knee OA patients. PRP contained a wide variety of cytokines, chemokines, growth factors, and autologous intra-articular PRP injection resulted in favorable outcomes in knee OA patients. Significant increases in levels of IL-1, IL-2, IL-7, IL-8, IL-9, IL-12, TNF-α, IL-17, PDGF-BB, bFGF, and MIP-1β were detected in PRP compared to PPP (*p* < 0.001). An in vitro study showed a marked increase in proliferation in OA chondrocytes cultured with PRP, compared to PPP and fetal bovine serum (*p* < 0.001). In a clinical study, knee OA patients treated with PRP showed improvement of physical function and pain, assessed by physical performance, Western Ontario and McMaster Universities Arthritis Index and visual analog scale. Our findings from both in vitro and clinical studies suggest that intra-articular PRP injection in knee OA patients may be a potential therapeutic strategy for alleviating knee pain and delaying the need for surgery.

## 1. Introduction

Knee osteoarthritis (OA) is a prevalent debilitating disease that results in healthcare burden worldwide [1]. Knee OA patients usually present with joint pain, swelling, tenderness and stiffness, which progressively worsens and hampers daily life activities. Given that knee OA is a chronic, insidious-onset disease that progresses at a variable pace [2], most knee OA patients have repeated flare ups, manifesting as morning stiffness, night pain, and joint swelling. The exact pathogenesis of knee OA remains unclear. However, it has been well-recognized that the pathological features of knee OA include articular cartilage degradation, formation of subchondral bone, and mild synovitis [3]. Although knee OA has been considered a simple wear and tear disease, mounting evidence has uncovered the role of inflammation in whole joint structures, both locally and systemically via several signaling pathways responsible for immune response [4,5]. Through an imbalance of catabolic and anabolic molecules in the cartilage matrix, the vicious cycle of inflammation leads to degradation of the articular cartilage [6]. Current interventions in the management of knee OA including physical and pharmacological therapies aim to relieve pain and improve physical function, but do not stop the disease progression. For that reason, in knee OA patients who are unable to tolerate oral pharmacological therapy, intra-articular injection has been suggested as an alternative therapeutic option for delaying surgical treatment [7,8,9].

Platelet-rich plasma (PRP), generally defined as an autologous platelet concentration, is a blood product with increased concentrations of platelets and is believed to contain a large amount of growth factors, chemokines, and cytokines via dense granules released by activated platelets [10,11]. In musculoskeletal regenerative medicine, PRP is a promising therapeutic tool for tendinopathy, muscle injury, bone fracture nonunion, and knee OA [12]. Various studies have shown encouraging outcomes for PRP used in OA; however, according to high heterogeneity methodology and lacking of standardization, PRP is provided as “uncertain” recommendation in OA research society international guideline [13] and “not recommend for or against” in American Academy of Orthopedic Surgeons clinical guidelines [14,15,16].

Given the debate surrounding use of PRP, we aimed to investigate concentrations of cytokines and growth factors in PRP compared to platelet-poor plasma (PPP) and determine not only the effects of PRP on migration, proliferation, and gene expression of human OA chondrocytes, but also the effect of intra-articular PRP injection in knee OA patients. We also hypothesized that increased levels of growth factors, cytokines, and chemokines in PRP may be responsible for the beneficial effects of PRP on knee OA in both clinical and in vitro studies.

## 2. Results

### 2.1. Patient Characteristics

In the present study, one patient was lost to follow-up, and one patient was undergone total knee arthroplasty (TKA). All 40 knee OA patients who met the criteria for analysis were female and categorized as Kellgren and Lawrence (KL) stage 2. Mean age of the patient was 67.1 ± 9.2 years, and mean body mass index (BMI) was 25.3 ± 3.7 kg/m^2^. Mean platelet count at baseline in the whole blood was 267.3 × 10^3^ ± 74.5 × 10^3^/μL; mean platelet count in PRP was 446.1 × 10^3^ ± 112.9 × 10^3^/μL. During the follow-up period, no complication was observed.

### 2.2. Growth Factor, Chemokine, and Cytokine Concentrations in PRP and PPP

In PRP derived from 40 knee OA patients, mean levels of inflammatory cytokines and growth factors including interleukin (IL)-1, IL-2, IL-7, IL-8, IL-9, IL-12, TNF-α, IL-17, PDGF-BB, and bFGF were found to be significantly higher than those in PPP (*p* < 0.001). In addition, IFN-γ, MIP-1β, VEGF, IP-10, and IL-5 levels in PRP were significantly greater than in PPP (*p* < 0.05). In contrast, there was no statistically significant difference in levels of anti-inflammatory cytokines including IL-4, IL-10, IL-13, and IL-1RA and other cytokines including IL-6, IL-15, MCP-1, MIP-1α, eotaxin, RANTES, G-CSF, and GM-CSF between PRP and PPP in knee OA patients. However, IL-10 levels in PRP were significantly lower than in PPP. Interestingly, RANTES, PDGF-BB and IP-10 were three cytokines with the highest concentrations in PRP, respectively (Figure 1).

### 2.3. Effect of PRP on Migration of OA Chondrocytes

Cell migration was evaluated using a scratch assay during a 24 and 48 h period in 3 distinct media conditions: 10% PRP, 10% PPP, and 10% fetal bovine serum (FBS) (control). Over 24 h and 48 h, OA chondrocytes cultured with 10% FBS showed significantly increased migration, as compared to those with 10% PPP (*p* < 0.021, *p* < 0.001, respectively). Instead, there was no significant difference in cell migration between OA chondrocytes with 10% FBS and with 10% PRP at 24 h but cell migration of OA chondrocytes with 10% FBS was higher than those with 10% PRP at 48 h (*p* < 0.01, Figure 2B). Over 48 h, OA chondrocytes cultured with 10% PRP showed significantly more migration than those with 10% PPP (*p* = 0.027) (Figure 2A,B).

### 2.4. Effect of PRP on Proliferation of OA Chondrocytes

OA chondrocytes cultured with 10% FBS showed significantly increased metabolic activity when compared to those with 10% PRP and 10% PPP over 3 days (*p* < 0.001), but no significant difference in metabolic activity of OA chondrocytes between 10% PRP and 10% PPP was observed (Figure 2C). Over 6 days of culture, metabolic activity of OA chondrocytes cultured with 10% PRP was significantly increased, compared to those with 10% FBS (*p* < 0.001) and 10% PPP (*p* < 0.001). Over a 9-day period, metabolic activity was considerably greater in chondrocytes cultured with 10% PRP than those with 10% FBS. (*p* < 0.001, Figure 2C).

### 2.5. Effect of PRP on Gene Expression of OA Chondrocytes

As depicted in Figure 3A, relative *SOX9* mRNA expression was detected to be significantly lower in 10% PPP than that in 10% FBS and 10% PRP over 3 days (*p* < 0.001), but significantly greater in 10% FBS than in 10% PRP and 10% PPP over 9 days (*p* < 0.01). Alternatively, there was no significant difference in relative *SOX9* mRNA expression among those three groups over 6 days. Consistent with this finding, compared with 10% PPP and 10% FBS, relative *COL2A1* mRNA expression was substantially enhanced in 10% PRP media over 6 and 9 days (Figure 3B). Conversely, relative aggrecan (*ACAN*) mRNA expression was shown to be significantly higher in OA chondrocytes cultured with 10% FBS than that those cultured with 10% PPP over 6 and 9 days (*p* < 0.001) (Figure 3C).

### 2.6. Clinical Evaluation after Intra-Articular PRP Injection in Knee OA Patients

Apart from an in vitro study demonstrating the beneficial effect of PRP on OA chondrocytes, clinical study was further conducted to investigate the effect of PRP on physical function and pain severity in knee OA patients. Overall physical performance, such as sit to stand (*p* < 0.001), time up and go (*p* < 0.001), and 3 min walk test (*p* < 0.001), significantly improved after 18-week intra-articular PRP injection as displayed in Figure 4. Clinical outcomes of knee OA patients were assessed using visual analog scale (VAS) and Western Ontario and McMaster Universities Arthritis Index (WOMAC) scores at 0- and 18-week interval. In knee OA patients treated with PRP, the VAS score used for determining pain severity was significantly decreased from 6.1 ± 0.4 to 2.4 ± 0.4 (*p* < 0.001), thereby indicating reduced knee pain in knee OA patients after intra-articular PRP injection. Besides this, intra-articular PRP injection significantly reduced pain and improved joint function in knee OA patients, in which the WOMAC score was used to evaluate pain and functional performance (*p* = 0.02) (Figure 5).

## 3. Discussion

Although TKA is the definitive management for severe OA patients, the pharmacological and procedural treatments are essential for whom are not surgical candidates. Due to the fact that existing understanding cannot fully explain the pathogenesis of knee OA, a number of studies have been conducted to overcome this obstacle. Based on the previous findings, biologic treatment is gaining popularity as a means of counteracting the molecular pathway responsible for knee OA pathology and delaying its progression [17]. Of various biologic treatments, intra-articular PRP injection is increasing interests as an alternative treatment modality for knee OA, which have shown superior clinical results to conventional treatment modalities [18,19,20,21,22,23]. Therefore, the current study attempts to determine concentrations of 27 cytokines in PRP and its effect on OA chondrocytes and in knee OA patients.

In this study, we investigated concentrations of cytokines, chemokines, and growth factors in both PRP and PPP of 40 knee OA patients using Bio-plex system based on a capture sandwich immunoassay. Among 27 cytokines, 8 inflammatory cytokines, 2 growth factors, and 1 chemokine were found to be significantly higher in PRP than those in PPP, consistent with previous studies [24]. As to the role of growth factors in knee OA, it has been shown that PDGF suppressed IL-1-induced chondrocyte inflammation [25]. By up-regulating other growth factors, such as transforming growth factor (TGF)-β, VEGF, or bone morphogenetic protein (BMP), bFGF has been reported to accelerate cartilage repair [26]. Given that inflammation is typically considered a localized protective response to vascularized injured tissues, the first phase of wound healing cascade is characterized by an influx of neutrophils and macrophages. These macrophages can influence the inflammation milieu by not only secreting several mediators including proinflammatory cytokines and chemokines, but also recruiting immune cells to sites of injury for tissue healing [27]. Through nuclear factor kappa b (NF-kB) signaling, a previous study demonstrated the transient proinflammatory activity followed by inflammation resolution [28]. In the light of the foregoing findings along with our results, given that PRP is composed of both anti-inflammatory cytokines and pro-inflammatory cytokines, it is tempting to speculate that a variety of proinflammatory cytokines and chemokines detected in PRP may be responsible for the clean-up signal complemented by anti-inflammation cytokines and growth factors with regenerative potential after resolution of inflammation.

Due to significant increases in levels of cytokines in PRP of knee OA patients, an in vitro study was further conducted to determine the beneficial effect of PRP against progressive knee OA in OA chondrocyte isolated from knee OA patients who underwent TKA. We observed that PRP had a greater effect on cell migration and proliferation of OA chondrocytes than PPP and FBS, which is in line with previous studies [29,30]. Moreover, a higher concentration of platelets in PRP showed more effects on cell migration [31]. We developed a manual centrifugation protocol instead of using kits for cost-saving reason; despite lower platelet yields, positive outcomes were observed in this clinical application. Therefore, we postulate that the PRP could influence the entire joint environment after intra-articular injection in knee OA patients regardless the effect of cell migration. Apart from investigating proliferation and migration of OA chondrocytes, the effect of PRP on redifferentiation of OA chondrocytes was further explored by determining expressions of cartilage-specific genes. Among various molecules known to be involved in cartilage matrix synthesis, COL2A1, a fundamental component of human hyaline cartilage, was degraded in knee OA [32]. In addition to COL2A1, SOX9 is a transcription factor, which is required for cartilage formation during embryogenesis [33] and for a production of collagen type II [34]. In this study, a substantial increase in relative *COL2A1* mRNA expression was detected in OA chondrocytes cultured with PRP, which is similar to an earlier study [35]. As the main proteoglycan in cartilage matrix, aggrecan is susceptible to proteolysis by aggrecanase or matrix metalloproteases (MMPs). One of important knee OA pathophysiology is an increased matrix degradation rate. According to our findings, it seems likely that the PRP treatment may have certain effect on alterations in expressions of aggrecan and *SOX9* in OA chondrocytes. Inconsistent with our finding above, a previous study demonstrated that PRP enhanced aggrecan mRNA expression in IL-1-induced meniscal cells after 10–14 days of culture [36]. To verify the contrasting result, we may need to culture OA chondrocytes with PRP treatment in a longer period of time to observe the effect of PRP on expressions of *ACAN* and *SOX9*. Also, in order to investigate the effect of PRP on redifferentiation of OA chondrocytes, further analyses on additional markers including *MMPs* or *COL1* are underway to elucidate the redifferentiation potential of OA chondrocytes.

In this study, clinical data and cytokine analysis were combined with an in vitro model to determine the effect of PRP on knee OA. In knee OA patients treated with intra-articular PRP injection, even though the VAS, WOMAC pain, and function scores were improved, the WOMAC stiffness score remained unchanged. This result supports several studies uncovering the effects of PRP on reducing pain and improving physical function [20,37,38]. However, it should be kept in mind that this is an observational study, in which none of control group was performed. Furthermore, the current research showed the favorable outcomes when PRP was used with a platelet concentration of 2-fold above baseline. A previous study suggested that a 5-fold increase in PRP above baseline was needed to achieve a favorable clinical outcome [39]. However, variability in dose, dosing interval, and duration of therapy is the main concern in clinical PRP usage. Additionally, the mechanisms underlying the protective effect of PRP against progressive knee OA have not yet been fully elucidated. Several studies along with this one support the notion that growth factors and cytokines were released via α-granules and PRP can stimulate the immune system via macrophage polarization [40,41]. PRP is an increased content of autologous platelets over the level in blood. A number of bioactive molecules are contained in the dense granules of platelets. Secreted proteins from PRP can be grouped in various families based on their biological activities. Factors such as PDGF, bFGF, VEGF, and several other chemokines and cytokines promote mitogenesis, chemotaxis, wound healing, and angiogenesis in an attempt to optimize the local environment of injured tissues [41]. These growth factors and cytokines initiate and enhance physiological processes that contribute to tissue recovery and healing after injury [41]. Previous study outlined that PRP induced diverse effects on articular chondrocytes in vitro, plausibly due to differences in the levels of platelets, leukocytes, growth factors, and other bioactive mediators [42]. We hypothesize that stimulation by the intra-articular injection of PRP could promote the production of cytokines from mesenchymal cells and inflammatory cells of cartilage tissues leading to the good clinical outcomes. Further studies will be needed to examine whether chondrocytes stimulated by PRP can express these factors or not and to determine the influence of PRP in in vivo animal model experiments.

Several caveats need to be mentioned in this study. Firstly, the redifferentiation potential of PRP on OA chondrocytes have not yet been completely explored in the present study because the chondrocytes usually lose their chondrogenic phenotypes in two-dimensional cultures [43]. Additional studies of PRP on OA chondrocytes in three-dimensional systems such as hydrogel or chondrocyte pellets are warranted to validate our findings. Secondly, a wide variation in the reported protocols for standardization and preparation of PRP may affect the platelet count in plasma. It seems that the reported protocol for PRP preparation utilized in our study produced lower-yield platelets [44]; however, our clinical study revealed that after intra-articular PRP injection, knee OA patients has significantly reduced knee pain and improved physical performance. We presume that PRP might influence an anabolic microenvironment, containing the suitable bioactive molecules, which contribute to maintaining the joint homeostasis, reducing pain and improving the articular function. Future in vitro and in vivo experiments are required to investigate the effects of PRP on therapeutic events in OA chondrocytes. Lastly, the relatively small sample size of this study limits the statistical power of our findings. In that context, the larger sample size and a longer follow-up period with standard methods are required to fully understand the PRP effects on knee OA.

The various PRP preparation methods lead to a high unpredictability of the product, resulting in inconsistent and doubtful outcomes. Recent study showed that hyperacute serum treatment exhibited a beneficial influence in relieving symptoms and providing an improvement in knee OA [45]. Hyperacute serum overcomes PRP disadvantage and includes a various composition in growth factors and cytokines with high potential, emerging as a promising therapeutic strategy and raising hope for future applications in OA.

In conclusion, this study investigated the comprehensive profiling of cytokines, chemokines, as well as growth factors in both PRP and PPP of knee OA subjects and determined their impact on OA chondrocytes in vitro and knee OA patients. Our findings indicated that the concentrations of several growth factors, cytokines, and chemokines in PRP were significantly higher than those in PPP. The addition of PRP further stimulated cell proliferation of OA chondrocytes with potentiated *SOX9* transcription resulting in sequentially elevated *COL2A1* and *ACAN* expression. In knee OA patients, the intra-articular PRP injection significantly reduced pain and improved physical function. We speculate that bioactive molecules in PRP might be an essential component in promoting tissue healing leading to favorable outcomes of knee OA patients. To ensure real-world applicability of PRP, the entire mechanisms underlying the beneficial effects of PRP against knee OA progression should be investigated, in addition to a standardized protocol for PRP preparation.

## 4. Materials and Methods

### 4.1. Subjects

A total of 40 knee OA patients defined by the American College of Rheumatology (ACR) criteria were recruited in this prospective observational study. All patients received radiographic evaluation of standing anteroposterior and lateral view of the knee and was further categorized as KL stages 1–2. Knee OA patients with symptomatic early-stage had a visual analogue score of 0–3 at rest and more than 4 in weight-bearing postures such as standing. Patients were included if they had discontinued any other pharmacological treatments in the preceding 3 months. The OA patients who had septic arthritis, rheumatoid arthritis, a history of knee surgery, significant trauma, or a musculoskeletal tumor were excluded from this study. Besides this, the patients who lost to follow-up and were treated for knee OA with other treatment modalities during PRP treatment meet the withdrawal criteria.

From June 2019 through May 2020, this study was conducted at a single institute. The study protocol abided by the 1975 Declaration of Helsinki ethical principles and was approved by the institutional review board of Faculty of Medicine, Chulalongkorn University, and written informed consent was obtained from each participant.

### 4.2. PRP Preparation

Peripheral blood was collected via venipuncture for 35 mL in citrate-phosphate-dextrose solution with adenine (CPDA) blood collection tube (Greiner Bio-One GmbH, Kremsmünster, Austria) from all patients. Due to the high cost of commercially available automated PRP kits, a manual double-spin preparation method was performed to prepare PRP as the following [43]. Whole blood was centrifuged at 1000× *g* revolutions per minute (rpm) for 10 min. Plasma and buffy coat were transferred to new tubes and centrifuged at 2000× *g* rpm for 10 min. The upper two-third layer was collected as PPP, while the lower one-third layer containing platelet pellet was collected as PRP. After activated using calcium chloride (CaCl_2_, Sigma-Aldrich, St. Louis, MO, USA), 5-mL of leukocyte-free autologous PRP was immediately injected into the patient’s knee. The PRP and PPP were stored at −80 °C for further in vitro evaluation.

### 4.3. Bio-Plex Pro Human Cytokine 27-Plex Assay

To measure concentrations of growth factors, chemokines, and cytokines, coupled magnetic beads were added into a 96-well plate followed by samples including PRP and PPP. Detection antibodies, standard reagents, and diluents were used in accordance with the manufacturers’ protocol. The 27-human cytokine concentrations including IL-1, IL-1 receptor antagonist (IL-1RA), IL-2, IL-4, IL-5, IL-6, IL-7, IL-8, IL-9, IL-10, IL-12, IL-13, IL-15, IL-17, granulocyte colony-stimulating factor (G-CSF), granulocyte-macrophage colony-stimulating factor (GM-CSF), eotaxin, RANTES (Regulated upon Activation, Normal T Cell Expressed and Presumably Secreted), interferon (IFN)-γ, tumor necrosis factor (TNF)-α, interferon γ-induced protein (IP)-10, monocyte chemoattractant protein (MCP)-1, macrophage inflammatory protein (MIP)-1α, basic fibroblast growth factor (bFGF), vascular endothelial growth factor (VEGF), and platelet derived growth factor (PDGF)-BB, were evaluated using the Bio-Plex 200 (Bio-Rad, Hercules, CA, USA).

### 4.4. Chondrocyte Isolation and Culture

Cartilage tissues were collected from the knees of 10 OA patients diagnosed by the ACR criteria at the time of surgery (Appendix A). After written informed consent was acquired, the human knee OA cartilage was obtained from surgical waste of patient undergoing TKA. The cartilage was rinsed with phosphate buffered saline (PBS) containing the following: 137 mM NaCl; 2.7 mM KCl; 10 mM Na_2_HPO_4_, and 1.8 mM KH_2_PO_4_ and minced into fine pieces. Enzymatic digestion with pronase (Sigma-Aldrich, St. Louis, MO, USA) and collagenase type II (Worthington Biochemical Corp., Lakewood, NJ, USA) and collagenase type II at temperature of 37 °C for 2 h was used to isolate chondrocytes from the articular cartilage. The digested articular cartilage was centrifuged at 1000× *g* rpm for 5 min, and supernatant was then discarded. Afterwards, the pellet was resuspended, seeded, and cultured in chondrogenic growth media (Dulbecco’s Modified Eagle Medium (DMEM; HyClone Laboratories, South Logan, UT, USA)/F12, 10% FBS (HyClone Laboratories, South Logan, UT, USA), 1% glutamax (GIBCO—BRL, Grand Island, NY, USA), 1% antibiotic-antimycotic (GIBCO—BRL, Grand Island, NY, USA), 0.05 mg/mL of ascorbic acid (Sigma-Aldrich, St. Louis, MO, USA). Culture media was replaced every 2–3 days, and knee OA chondrocyte passages 1–3 were utilized in the subsequent experiments.

### 4.5. Effect of PRP on OA Chondrocyte Migration

Chondrocyte migration was evaluated using scratch assay. Chondrocytes were seeded at a density of 2.5 × 10^5^ cells/mL into 6-well plates at 37 °C, 5% CO_2_. Control media (10% FBS) was used until cells reached 90% confluence. The pipette tip was used for scratching. Following that, media containing 10% PRP, 10% PPP, and 10% FBS (control) were added. PRP and PPP used in the in vitro experiments were pooled from all donors. PRP and PPP contained 2 U/mL heparin (Sigma-Aldrich, St. Louis, MO, USA) to prevent fibrin clot. The gaps were measured and taken by light microscope with digital camera at 0, 24, and 48 h.

### 4.6. Effect of PRP on OA Chondrocyte Proliferation

The effect of PRP on chondrocyte proliferation was evaluated for metabolic activity using the methyl thiazolyl tetrazolium (MTT) assay (C_18_H_16_BrN_5_S; Bio Basic Inc., Amherst, NY, USA) in 3 different media conditions: 10% PRP, 10% PPP, and 10% FBS (control) for 3 time points’ period: 3, 6, and 9 days of culture. Knee OA chondrocytes were seeded at 1000 cells into 96-well plates in control media (10% FBS). After 48 h, the media was removed, and the chondrocytes were treated within different experimental media: 10% PRP, 10% PPP, and 10% FBS (control). Chondrocyte viability was measured in triplicate wells per condition at 3, 6 and 9 days using at 570 nm absorbance of reduced MTT.

### 4.7. Effect of PRP on Gene Expression of OA Chondrocytes

Expressions of chondrogenic-specific genes including *SOX9*, *COL2A1*, and *ACAN* (aggrecan) were determined using quantitative real-time polymerase chain reaction (PCR). PCR. 2.5 × 10^5^ cells/mL OA chondrocytes were seeded in 6-well plates and cultured in control media for 48 h. The media was changed to three different conditions: 10% PRP, 10% PPP, and 10% FBS (control). Total mRNA was extracted using Trizol reagent Invitrogen, Carlsbad, CA, USA). The extracted mRNA was reverse transcribed with a cDNA synthesis kit (Bioline/Meridian Bioscience, Luckenwalde, Germany). The chondrogenic-specific genes including SRY-box transcription factor (*SOX9;* 5′-atctgaagaaggagagcgag-3′ and 5′-tcagaagtctcccagagcttg-3′), collagen type II alpha 1 (*COL2A1*; 5′-ctggctcccaacactgccaacgtc-3′ and 5′-tcctttgggtttgcaacggattgt-3′), and aggrecan (*ACAN*; 5′-tgaggagggctggaacaagtacc-3′ and 5′-ggaggtggtaattgcagggaaca-3′) were examined at days 3, 6, and 9 using real-time PCR performed in a thermal cycler (Applied Biosystems, Inc., Foster City, CA, USA) (Appendix A). Glyceraldehyde 3-phosphate dehydrogenase (*GADPH*; 5′-ttccattgacctcaactacat-3′ and 5′-gaggggccatccacagtctt-3′) expression was used as an external reference gene, and relative mRNA expression for each target gene was calculated using 2^−ΔΔCt^ method.

### 4.8. Clinical Evaluation after Intra-Articular PRP Injection in Knee OA Patients

Intra-articular injection was aseptically performed by a single orthopedic surgeon who was not involved in the assessment. On the first outpatient department visit and at 6 weeks, PRP activated with CaCl_2_ was immediately injected into the knee joint via the lateral mid-patellar approach. Prior to treatment and after starting treatment at 6, 12, and 18 weeks, physical function and pain were both evaluated using standardized questionnaires: WOMAC and VAS.

### 4.9. Statistical Analysis

All statistical analyses were performed using SPSS version 22.0 (SPSS Inc., Chicago, IL, USA). All data are presented as the mean ± standard deviation (SD). Statistically significant differences in normal distributed data between two dependent groups were executed using a paired Student *t*-test. Whether there are any statistically significant differences between the means of three or more independent groups was determined using the one-way analysis of variance (ANOVA) with Tukey’s post hoc test. For all analyses, statistical significance was set at *p* < 0.05.

## Figures and Tables

**Figure 1 ijms-23-00890-f001:**
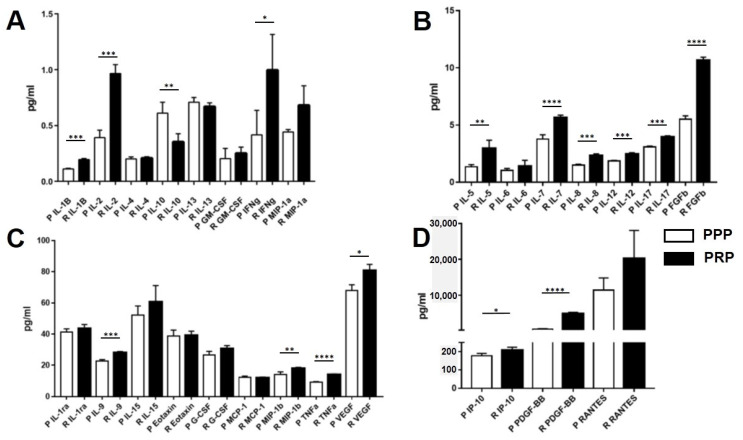
Profiles of cytokines, chemokines, and growth factors in PRP and PPP of knee OA patients at various concentrations: (**A**) 0–1.5 pg/mL; (**B**) 1.5–15 pg/mL; (**C**) 15–80 pg/mL; (**D**) 100–30,000 pg/mL. Abbreviations: OA, osteoarthritis; PRP, platelet-rich plasma; PPP, platelet-poor plasma. * *p* < 0.05, ** *p* < 0.01, *** *p* < 0.001, **** *p* < 0.0001.

**Figure 2 ijms-23-00890-f002:**
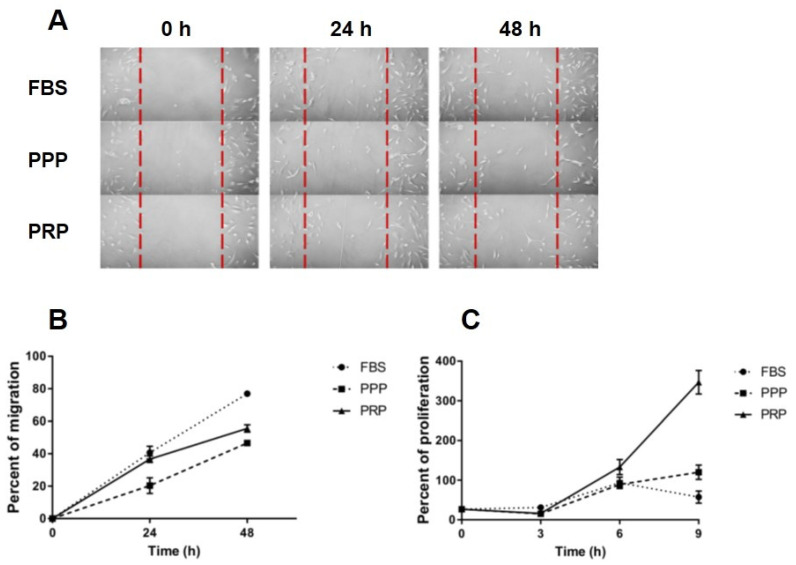
Effects of PRP, PPP, and FBS on migration and proliferation of OA chondrocytes: (**A**) scratch assay images captured at 0, 24, and 48 h using an inverted light microscope to observe migration of knee OA chondrocytes; (**B**) migration of knee OA chondrocytes cultured with PRP, PPP, and FBS at 0, 24, and 48 h; (**C**) proliferation of knee OA chondrocytes cultured with PRP, PPP and FBS at 0, 3, 6, and 9 h. Abbreviations: FBS, fetal bovine serum; OA, osteoarthritis; PRP, platelet-rich plasma; PPP, platelet-poor plasma.

**Figure 3 ijms-23-00890-f003:**
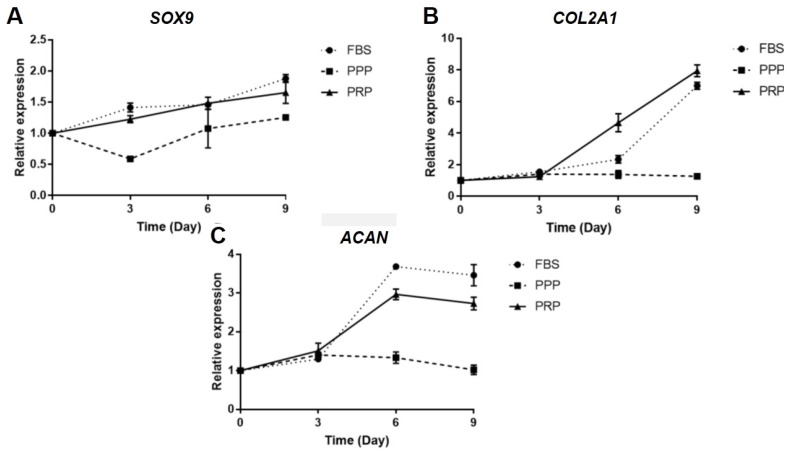
Effects of PRP, PPP, and FBS on expressions of cartilage-specific genes in knee OA chondrocytes: (**A**) relative *SOX9* mRNA expression; (**B**) relative *COL2A1* mRNA expression; (**C**) relative *ACAN* mRNA expression. Abbreviations: *ACAN*, aggrecan; *COL2A1*, collagen type II alpha 1; FBS, fetal bovine serum; OA, osteoarthritis; PRP, platelet-rich plasma; PPP, platelet-poor plasma; *SOX9*, SRY-box transcription factor.

**Figure 4 ijms-23-00890-f004:**
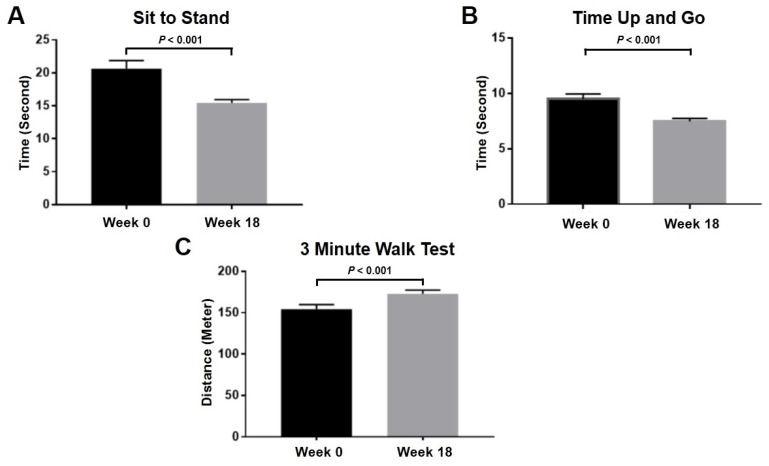
Comparison of physical performance at 0 and 18 weeks following intra-articular PRP injection in knee OA patients: (**A**) sit to stand; (**B**) time up to go; (**C**) 3 min walk test.

**Figure 5 ijms-23-00890-f005:**
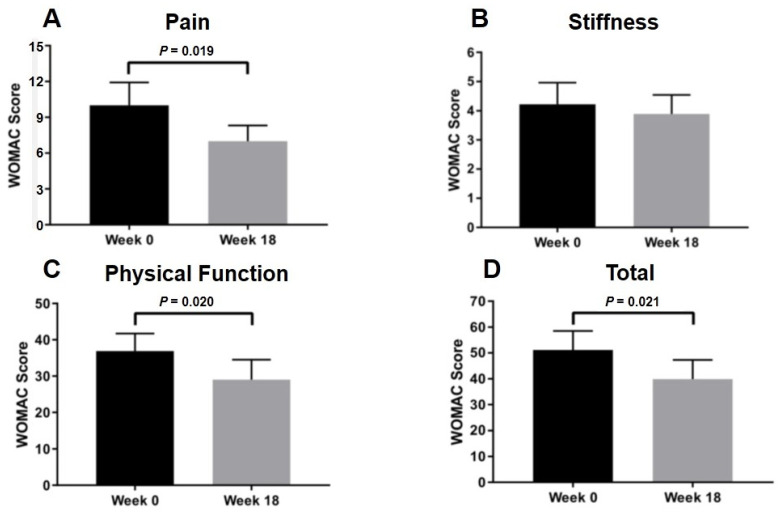
Comparison of WOMAC score at 0 and 18 weeks following intra-articular PRP injection in knee OA patients: (**A**) pain; (**B**) stiffness; (**C**) physical function; (**D**) total score. Abbreviations: WOMAC, Western Ontario and McMaster Universities Osteoarthritis.

## Data Availability

The data presented in this study are available on request from the corresponding authors.

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
