# Peer review of "Cytokine Profiling and Intra-Articular Injection of Autologous Platelet-Rich Plasma in Knee Osteoarthritis"

_ijms, 2022, doi:10.3390/ijms23020890_

Round 1

Reviewer 1 Report

The authors examined effect of intra-articular PRP injection on chondrocytes from the human knee OA cartilage and clinical study results of OA patients. From their results, high concentration of growth factors, cytokines, and chemokines in PRP can be an essential component in promoting tissue healing. This study is interesting and the message can advantage the treatment strategy of OA. However, there are some issues to be addressed in this manuscript. 

1) It is hard to believe that high concentration of growth factors, cytokines, and chemokines are derived only from injected PRP. Is that true ? Usually, stimulation by the injection of PRP can promote production of cytokines in mesenchymal and inflammatory cells of cartilage tissues. The authors must examine whether chondrocytes stimulated by PRP can express or  produce those factors or not. 

2) In this manuscript, the authors did not explain how factors detected by the injection lead to the good clinical results. Additional in vitro experiments are needed to examine effects of PRP on therapeutic events in OA chondrocytes. 

Author Response

Thank you very much for your mail and the reviewers’ comments on our manuscript entitled “Cytokine Profiling and Intra-articular Injection of Autologous Platelet-Rich Plasma in Knee Osteoarthritis”. We have revised accordingly for all the reviewers’ comments. We also have done following on the check list.

Comments from the editors and reviewers: 

Reviewer #1:

The authors examined effect of intra-articular PRP injection on chondrocytes from the human knee OA cartilage and clinical study results of OA patients. From their results, high concentration of growth factors, cytokines, and chemokines in PRP can be an essential component in promoting tissue healing. This study is interesting and the message can advantage the treatment strategy of OA. However, there are some issues to be addressed in this manuscript. 

1) It is hard to believe that high concentration of growth factors, cytokines, and chemokines are derived only from injected PRP. Is that true? Usually, stimulation by the injection of PRP can promote production of cytokines in mesenchymal and inflammatory cells of cartilage tissues. The authors must examine whether chondrocytes stimulated by PRP can express or produce those factors or not. 

Response: Thank you very much for valuable suggestion. We have addressed that point in Discussion:

PRP is an increased content of autologous platelets over the level in blood. A number of bioactive molecules are contained in the dense granules of platelets. Secreted proteins from PRP can be grouped in various families based on their biological activity. Factors such as PDGF, bFGF, VEGF, and several other chemokines and cytokines promote mitogenesis, chemotaxis, wound healing, and angiogenesis in an attempt to optimize the local environment of injured tissues. These growth factors and cytokines initiate and enhance physiological processes that contribute to tissue recovery and healing after injury. Moreover, stimulation by the intra-articular injection of PRP could promote the production of cytokines in mesenchymal cells and inflammatory cells of cartilage tissues leading to the good clinical outcomes. Further studies will be needed to examine whether chondrocytes stimulated by PRP can express or produce these factors or not.”

2) In this manuscript, the authors did not explain how factors detected by the injection lead to the good clinical results. Additional in vitro experiments are needed to examine effects of PRP on therapeutic events in OA chondrocytes. 

Response: We have stated that point in Discussion:

Secondly, a wide variation in the reported protocols for standardization and preparation of PRP may affect the platelet count in plasma. It seems that the reported protocol for PRP preparation utilized in our study produced lower-yield platelets [43]. Despite this, our clinical study revealed that after intra-articular PRP injection, knee OA patients has significantly reduced knee pain and improved physical performance. Future in vitro experiments are required to investigate the effects of PRP on therapeutic events in OA chondrocytes.

We agree with the entire reviewers’ comments. We hope that all revisions will meet all the reviewers’ recommendation.

Should you have any queries, please do not hesitate to contact me. Along with this letter, we have enclosed the revised manuscript with all changes marked.

Once again, we would like to thank you very much in advance and we are looking forward to hearing from you.

Sincerely yours,

Sittisak Honsawek, MD, PhD

Department of Biochemistry,

Osteoarthritis and Musculoskeleton Research Unit

Faculty of Medicine, Chulalongkorn University,

Bangkok 10330 Thailand.

Tel.  (662) 256-4482 Fax. (662) 256-4482

E-Mail : sittisak.h@chula.ac.th

Reviewer 2 Report

Dear Authors,

the present Manuscript "Cytokine Profiling and Intra-articular Injection of Autologous 2 Platelet-Rich Plasma in Knee Osteoarthritis"  aimed to examine cytokine profiling in both (PRP) and platelet-poor plasma (PPP) of knee OA patients and to determine the effects of PRP on OA chondrocytes and knee OA patients.

I think that the in vitro results shown are not sufficient to support the discussion and are not well described. 

The in vitro study does not evidence any induction on differentiation of chondrocyte in presence of PRP, probably because you should use a three-dimensional (3D) system such as Hydrogel.

I suggest repeating the experiment using a 3 D system.

Best regards

Author Response

Thank you very much for your mail and the reviewers’ comments on our manuscript entitled “Cytokine Profiling and Intra-articular Injection of Autologous Platelet-Rich Plasma in Knee Osteoarthritis”. We have revised accordingly for all the reviewers’ comments. We also have done following on the check list.

Comments from the editors and reviewers: 

Reviewer #2:

the present Manuscript "Cytokine Profiling and Intra-articular Injection of Autologous 2 Platelet-Rich Plasma in Knee Osteoarthritis" aimed to examine cytokine profiling in both (PRP) and platelet-poor plasma (PPP) of knee OA patients and to determine the effects of PRP on OA chondrocytes and knee OA patients.

I think that the in vitro results shown are not sufficient to support the discussion and are not well described. 

The in vitro study does not evidence any induction on differentiation of chondrocyte in presence of PRP, probably because you should use a three-dimensional (3D) system such as Hydrogel.

I suggest repeating the experiment using a 3 D system.

Response: Thank you very much for valuable suggestion. We have addressed that point in Discussion:

“We observed that PRP had a greater effect on cell migration and proliferation of OA chondrocytes than PPP and FBS, which is in line with previous studies [29, 30]. Moreover, higher concentration of platelets in PRP showed more effects on cell migration [31]. We developed manual centrifugation protocol instead of using kits for cost-saving reason; despite lower platelet yields, positive outcomes were observed in this clinical application. Therefore, the PRP could influence the entire joint environment after intra-articular injection in knee OA patients regardless the effect of cells migration.”

“To investigate the effect of PRP on redifferentiation of OA chondrocytes, further analyses on additional markers including MMPs or COL1 are underway to elucidate the redifferentiation potential of OA chondrocytes.”

“PRP is an increased content of autologous platelets over the level in blood. A number of bioactive molecules are contained in the dense granules of platelets. Secreted proteins from PRP can be grouped in various families based on their biological activity. Factors such as PDGF, bFGF, VEGF, and several other chemokines and cytokines promote mitogenesis, chemotaxis, wound healing, and angiogenesis in an attempt to optimize the local environment of injured tissues. These growth factors and cytokines initiate and enhance physiological processes that contribute to tissue recovery and healing after injury. Moreover, stimulation by the intra-articular injection of PRP could promote the production of cytokines in mesenchymal cells and inflammatory cells of cartilage tissues leading to the good clinical outcomes. Further studies will be needed to examine whether chondrocytes stimulated by PRP can express or produce these factors or not.”

Several caveats need to be mentioned in this study. Firstly, the redifferentiation potential of PRP on OA chondrocytes have not yet been completely explored in the present study, because the chondrocytes usually lose their chondrogenic phenotype in two-dimensional culture [42]. Additional studies of PRP on OA chondrocytes in three-dimensional system such as hydrogel or chondrocyte pellets are warranted to validate our findings.”

We agree with the entire reviewers’ comments. We hope that all revisions will meet all the reviewers’ recommendation.

Should you have any queries, please do not hesitate to contact me. Along with this letter, we have enclosed the revised manuscript with all changes marked.

Once again, we would like to thank you very much in advance and we are looking forward to hearing from you.

Sincerely yours,

Sittisak Honsawek, MD, PhD

Department of Biochemistry,

Osteoarthritis and Musculoskeleton Research Unit

Faculty of Medicine, Chulalongkorn University,

Bangkok 10330 Thailand.

Tel.  (662) 256-4482 Fax. (662) 256-4482

E-Mail : sittisak.h@chula.ac.th

Reviewer 3 Report

This study is interesting becuase it investigates the cytokine content in PRP and both in vitro and in vivo evaluation of the (same?) obtained PRP. However, my concern is that the PLT concentration used in this study was LOW, therefore it doesn't really help to answer the question about the most effective concentration of PLT or cytokines for OA treatment. From the clinical point of view, this study doesn't present anything new.

"Several caveats need to be mentioned in this study. Firstly, the redifferentiation potential of PRP on OA chondrocytes have not yet been completely explored in the present study, because the chondrocytes usually lose their chondrogenic phenotype in 2D culture [40]."

Why did you use the 2D cell culture model for OA chondrocytes in your study? Many publications showed already that 3D models (even the simple ones like chondrocyte pellets) are more suitable for OA investigation. Also to investigate the redifferentiation of OA chondrocytes only the expression of COL2, SOX9 and ACAN were measured. Analysis should also include some additional markers like MMPs or osteogenic markers, especially COL1. These results are not enough to consider the redifferentiation potential.

"We observed that PRP had a greater effect on cell migration and proliferation of OA chondrocytes than PPP and FBS, which is in line with previous studies [26, 27]. However, the effect of PRP on chondrocyte migration was not obvious in this study. Recent study unveiled that PRP significantly promoted cell migration, especially at higher concentrations [28]. The explanation for this finding may be attributable to low platelet concentrations in PRP utilized in our study."

Please rewrite this fragment, because it is not clear if in your study the results from scratch assay were satisfactory or not. Also, based on the current knowledge about PRP that the platelet concentration is critical for its clinical effect. It is hard to find this information in your paper and also I would like to see some arguments why this LOW concentration was used in the study.

The manuscript is missing a lot of information regarding the devices and chemicals which were used during this study:

  • What kind of tubes or kits were used to collect blood? Was it a brand-name PRP isolation kit or generic tubes? Please describe in detail as it may help comparison with other published data. 
  • Was this PRP leukocyte free or leukocyte rich? What was the mean number of WBC and RBC in the whole patient group? Same for PPP (also if any PLT were still present)
  • Please describe how long was the PRP prepared before the intra-articular injection. Did patients receive their own (autologous) PRP?
  • What was the injected volume per knee? 
  • How was the PRP used in in vitro experiments - was it pooled from all donors? Were there any anticoagulants used during the experiments?
  • Figure 1: in the graphs, it is not clear what are “P” and “R” before the names of the cytokines. It should not be necessary to have those letters if the labels on the right are correctly written. Moreover, please stick to one name, either “plasma”as stated in the colour label or “PPP” as written in the figure legend. Statistically significant expression should be labelled as stated in the figure with the sign *.
  • For your work to be replicated/ validated by others, please provide more detailed information about chemicals that were used

            company, city, country

            for kits: commercial name or cat number

            primers information: sequences

  • What kind of ANOVA was used in statistical analysis?

Need to be added:

Line 67: “despite the fact that previous studies have shown encouraging outcomes with intraarticular PRP injections”. This information is incomplete, as other studies proof the opposite, as explained later on in lines 69-70, probably due to the lack of standarization. Please check literature:

Roffi A.; di Matteo B. Platelet-rich plasma for the treatment of bone defects: from pre-clinical rational to evidence in the clinical practice. A systematic review. Intern orthop. 2017; 41(2): p. 221-237.

Mariani E.; Canella V. Leukocyte-rich platelet-rich plasma injections do not up-modulate intra-articular pro-inflammatory cytokines in the osteoarthritic knee. PloS one. 2016; 11(6).

Filardo G.; Kon E. PRP: product rich in placebo? Knee Surg. Sports Traumatol. Arthrosc. 2014; 24(12): p. 3702-3703.

Campbell K.A.; Saltzman B.M. Does intra-articular platelet-rich plasma injection provide clinically superior outcomes compared with other therapies in the treatment of knee osteoarthritis? A systematic review of overlapping meta-analyses. Arthroscopy. 2015; 31(11): p. 2213-2221.

Section 2.3 Is 10% PRP significantly different as 10% FBS after 48 hours?

Line 138, is 10% FBS also significantly different to 10% PRP? Please add P-value if important.

Section 4.4, patient information in this section is needed.

An important issue has not been discussed, a similar study with a blood derivative that overcomes PRP disadvantages was performed and may be of the interest of your discussion.

Olmos Calvo I.; Fodor E.; A pilot clinical study of hyperacute serum treatment in osteoarthritic knee joint: cytokine changes and clinical effects. Curr. Issues Mol. Biol. 2021.

Further discussion about the molecules that were significantly up- or down-regulated should be included.

Typos

Line 7: wrong spelling “nd esearch”

Line 17: extra spacing

Line 18: extra spacing

Line 18: wrong spelling “allied”

Line 23: missing ending bracket

Line 32: extra “and” before “growth factors”

Line 79-86: this information belongs to “materials and methods”

Lines 102 and 103 may be part of the section “materials and methods”.

Lines 108: “significantly more migration than” has another size and italics format.

Line 113. “PC” was not explained before.

Lines 117 and 118 may be part of the section “materials and methods”.

Line 125, can “PRP” be where “PPP” is written? Please check.

Lines 128 and 129 may be part of the section “materials and methods”.

Line 160, “abbreviations:” should be added or deleted.

Lines 167 and 168 need re-writing. The sentence is not correct.

Lines 207 and 208, “influence the entire joint environment” is in grey color and italics format.

Line 219, “metalloprotease” needs to be plural, and the abbreviation, MMPs.

Line 343, what does “(AT)” mean?

Line 344, what does “PC” mean?

Author Response

December 23, 2021

Ms. Fennie Fang,
Assistant Editor, MDPI

Journal: International Journal of Molecular Sciences
Manuscript ID: ijms-1516487
Title: Cytokine Profiling and Intra-articular Injection of Autologous Platelet-Rich Plasma in Knee Osteoarthritis

Dear Editor,

Thank you very much for your mail and the reviewers’ comments on our manuscript entitled “Cytokine Profiling and Intra-articular Injection of Autologous Platelet-Rich Plasma in Knee Osteoarthritis”. We have revised accordingly for all the reviewers’ comments. We also have done following on the check list.

Comments from the editors and reviewers: 

Reviewer #3:

This study is interesting becuase it investigates the cytokine content in PRP and both in vitro and in vivo evaluation of the (same?) obtained PRP. However, my concern is that the PLT concentration used in this study was LOW, therefore it doesn't really help to answer the question about the most effective concentration of PLT or cytokines for OA treatment. From the clinical point of view, this study doesn't present anything new.

"Several caveats need to be mentioned in this study. Firstly, the redifferentiation potential of PRP on OA chondrocytes have not yet been completely explored in the present study, because the chondrocytes usually lose their chondrogenic phenotype in 2D culture [40]."

Why did you use the 2D cell culture model for OA chondrocytes in your study? Many publications showed already that 3D models (even the simple ones like chondrocyte pellets) are more suitable for OA investigation. Also to investigate the redifferentiation of OA chondrocytes only the expression of COL2, SOX9 and ACAN were measured. Analysis should also include some additional markers like MMPs or osteogenic markers, especially COL1. These results are not enough to consider the redifferentiation potential.

Response: Thank you very much for valuable suggestion. We have addressed that point in Discussion:

Several caveats need to be mentioned in this study. Firstly, the redifferentiation potential of PRP on OA chondrocytes have not yet been completely explored in the present study, because the chondrocytes usually lose their chondrogenic phenotype in two-dimensional culture [42]. Additional studies of PRP on OA chondrocytes in three-dimensional system such as hydrogel or chondrocyte pellets are warranted to validate our findings.”

“To investigate the effect of PRP on redifferentiation of OA chondrocytes, further analyses on additional markers including MMPs or COL1 are underway to elucidate the redifferentiation potential of OA chondrocytes.”

"We observed that PRP had a greater effect on cell migration and proliferation of OA chondrocytes than PPP and FBS, which is in line with previous studies [26, 27]. However, the effect of PRP on chondrocyte migration was not obvious in this study. Recent study unveiled that PRP significantly promoted cell migration, especially at higher concentrations [28]. The explanation for this finding may be attributable to low platelet concentrations in PRP utilized in our study."

Please rewrite this fragment, because it is not clear if in your study the results from scratch assay were satisfactory or not. Also, based on the current knowledge about PRP that the platelet concentration is critical for its clinical effect. It is hard to find this information in your paper and also I would like to see some arguments why this LOW concentration was used in the study.

Response: We rewrote and stated that point in Discussion:

“We observed that PRP had a greater effect on cell migration and proliferation of OA chondrocytes than PPP and FBS, which is in line with previous studies [29, 30]. Moreover, higher concentration of platelets in PRP showed more effects on cell migration [31]. We developed manual centrifugation protocol instead of using kits for cost-saving reason; despite lower platelet yields, positive outcomes were observed in this clinical application. Therefore, the PRP could influence the entire joint environment after intra-articular injection in knee OA patients regardless the effect of cells migration.”

The manuscript is missing a lot of information regarding the devices and chemicals which were used during this study:

  • What kind of tubes or kits were used to collect blood? Was it a brand-name PRP isolation kit or generic tubes? Please describe in detail as it may help comparison with other published data. 

Response: We included that point in Materials and Methods:

“Peripheral blood was collected via venipuncture for 35 mL in citrate-phosphate-dextrose solution with adenine (CPDA) blood collection tube (Greiner Bio-One GmbH, Kremsmünster, Austria) from all patients. Due to the high cost of commercially available  automated PRP kits, a manual double-spin preparation method was performed to prepare PRP as the following [43].”

  • Was this PRP leukocyte free or leukocyte rich? What was the mean number of WBC and RBC in the whole patient group? Same for PPP (also if any PLT were still present)
  • Please describe how long was the PRP prepared before the intra-articular injection. Did patients receive their own (autologous) PRP?
  • What was the injected volume per knee? 

Response: We stated those points in Materials and Methods:

After activated using calcium chloride (CaCl2, Sigma-Aldrich, St. Louis, MO, US)), 5-mL of leukocyte-free autologous PRP was immediately injected into the patient’s knee.

  • How was the PRP used in in vitro experiments - was it pooled from all donors? Were there any anticoagulants used during the experiments?

Response: We addressed that point in Materials and Methods:

“PRP and PPP used in the in vitro experiments were pooled from all donors. PRP and PPP contained 2 U/mL heparin (Sigma-Aldrich) to prevent fibrin clot.”

  • Figure 1: in the graphs, it is not clear what are “P” and “R” before the names of the cytokines. It should not be necessary to have those letters if the labels on the right are correctly written. Moreover, please stick to one name, either “plasma”as stated in the colour label or “PPP” as written in the figure legend. Statistically significant expression should be labelled as stated in the figure with the sign *.

Response: We revised Figure 1 and labelled all statistically significant expression in the figure 1 with the sign “ * ”.

Figure 1. Profiles of cytokines, chemokines, and growth factors in PRP and PPP of knee OA patients at various concentrations. (A) 0-1.5 pg/mL. (B) 1.5-15 pg/mL. (C) 15-80 pg/mL. (D) 100-30,000 pg/mL. Abbreviations: OA, osteoarthritis; PRP, platelet-rich plasma; PPP, platelet-poor plasma. *P<0.05, **P<0.01, ***P<0.001, ****P<0.0001.

  • For your work to be replicated/ validated by others, please provide more detailed information about chemicals that were used

            company, city, country

            for kits: commercial name or cat number

            primers information: sequences

  • What kind of ANOVA was used in statistical analysis?

Need to be added:

Response: We provided the needed details in Materials and Methods (section 4.2 – 4.9).

Line 67: “despite the fact that previous studies have shown encouraging outcomes with intraarticular PRP injections”. This information is incomplete, as other studies proof the opposite, as explained later on in lines 69-70, probably due to the lack of standarization. Please check literature:

Roffi A.; di Matteo B. Platelet-rich plasma for the treatment of bone defects: from pre-clinical rational to evidence in the clinical practice. A systematic review. Intern orthop. 2017; 41(2): p. 221-237.

Mariani E.; Canella V. Leukocyte-rich platelet-rich plasma injections do not up-modulate intra-articular pro-inflammatory cytokines in the osteoarthritic knee. PloS one. 2016; 11(6).

Filardo G.; Kon E. PRP: product rich in placebo? Knee Surg. Sports Traumatol. Arthrosc. 2014; 24(12): p. 3702-3703.

Campbell K.A.; Saltzman B.M. Does intra-articular platelet-rich plasma injection provide clinically superior outcomes compared with other therapies in the treatment of knee osteoarthritis? A systematic review of overlapping meta-analyses. Arthroscopy. 2015; 31(11): p. 2213-2221.

Response: We revised in the introduction and included 3 references:

Various studies have shown encouraging outcomes for PRP used in OA; however, according to high heterogeneity methodology and lacking of standardization, PRP is provided as “uncertain” recommendation in OA research society international guideline [13] and “not recommend for or against” in American Academy of Orthopaedic Surgeons clinical guidelines [14-16].

13. McAlindon TE, Bannuru RR, Sullivan MC, Arden NK, Berenbaum F, Bierma-Zeinstra SM, et al. OARSI guidelines for the non-surgical management of knee osteoarthritis. Osteoarthritis Cartilage, 2014;22(3):363-388.

  1. Chen P, Huang L, Ma Y, Zhang D, Zhang X, Zhou J, et al. Intra-articular platelet-rich plasma injection for knee osteoarthritis: a summary of meta-analyses. J Orthop Surg Res, 2019;14(1):385.
  2. Jevsevar DS, et al. The American Academy of Orthopaedic Surgeons evidence-based guideline on: treatment of osteoarthritis of the knee, 2nd edition. J Bone Joint Surg Am, 2013;95(20):1885-1886.

Section 2.3 Is 10% PRP significantly different as 10% FBS after 48 hours?

Response: Yes, it is. We stated that in the results, section2.3:

but cell migration of OA chondrocytes with 10% FBS was higher than those with 10% PRP at 48 hours (P<0.01, Figure 2B).

Line 138, is 10% FBS also significantly different to 10% PRP? Please add P-value if important.

Response: We added P-value in the results, section 2.5.

Section 4.4, patient information in this section is needed.

Response: We provided the subject information in the section 4.4 and supplement Table S1:

Cartilage tissues were collected from the knees of 10 OA patients diagnosed by the ACR criteria at the time of surgery (Table S1).

An important issue has not been discussed, a similar study with a blood derivative that overcomes PRP disadvantages was performed and may be of the interest of your discussion.

Olmos Calvo I.; Fodor E.; A pilot clinical study of hyperacute serum treatment in osteoarthritic knee joint: cytokine changes and clinical effects. Curr. Issues Mol. Biol. 2021.

Response: We have discussed that point and included in Discussion and References:

“The various PRP preparation methods lead to a high unpredictability of the product, resulting in inconsistent and doubtful outcomes. Recent study showed that hyperacute serum treatment exhibited a beneficial influence in relieving symptoms and providing an improvement in knee OA [44]. Hyperacute serum overcomes PRP disadvantage and includes a various composition in growth factors and cytokines with high potential, emerging as a promising therapeutic strategy and raising hope for future applications in OA.”

Further discussion about the molecules that were significantly up- or down-regulated should be included.

Response: We added more details in Discussion:

“PRP is an increased content of autologous platelets over the level in blood. A number of bioactive molecules are contained in the dense granules of platelets. Secreted proteins from PRP can be grouped in various families based on their biological activity. Factors such as PDGF, bFGF, VEGF, and several other chemokines and cytokines promote mitogenesis, chemotaxis, wound healing, and angiogenesis in an attempt to optimize the local environment of injured tissues. These growth factors and cytokines initiate and enhance physiological processes that contribute to tissue recovery and healing after injury. Moreover, stimulation by the intra-articular injection of PRP could promote the production of cytokines in mesenchymal cells and inflammatory cells of cartilage tissues leading to the good clinical outcomes. Further studies will be needed to examine whether chondrocytes stimulated by PRP can express or produce these factors or not.”

Typos Line 7: wrong spelling “nd esearch”

Response: We have already corrected that.

Line 17: extra spacing

Line 18: extra spacing

Line 18: wrong spelling “allied”

Line 23: missing ending bracket

Line 32: extra “and” before “growth factors”

Response: We have already corrected all of those.

Line 79-86: this information belongs to “materials and methods”

Response: We have already moved to Materials and Methods:

Lines 102 and 103 may be part of the section “materials and methods”.

Response: We have already moved to Materials and Methods:

Lines 108: “significantly more migration than” has another size and italics format.

Response: We have already corrected that.

Line 113. “PC” was not explained before.

Response: We deleted it.

Lines 117 and 118 may be part of the section “materials and methods”.

Response: We have already moved to Materials and Methods:

Line 125, can “PRP” be where “PPP” is written? Please check.

Response: We have checked and corrected that. Thank you.

Lines 128 and 129 may be part of the section “materials and methods”.

Response: We have already moved to Materials and Methods:

Line 160, “abbreviations:” should be added or deleted.

Response: We deleted it.

Lines 167 and 168 need re-writing. The sentence is not correct.

Response: We revised that sentence.

“Although TKA is the definitive management for severe OA patients, the pharmacological and procedural treatments are essential for whom are not surgical candidates.”

Lines 207 and 208, “influence the entire joint environment” is in grey color and italics format.

Response: We have already corrected that.

Line 219, “metalloprotease” needs to be plural, and the abbreviation, MMPs.

Response: We have already corrected that.

Line 343, what does “(AT)” mean?

Response: We deleted it.

Line 344, what does “PC” mean?

Response: We have already corrected that.

We agree with the entire reviewers’ comments. We hope that all revisions will meet all the reviewers’ recommendation.

Should you have any queries, please do not hesitate to contact me. Along with this letter, we have enclosed the revised manuscript with all changes marked.

Once again, we would like to thank you very much in advance and we are looking forward to hearing from you.

Sincerely yours,

Sittisak Honsawek, MD, PhD

Department of Biochemistry,

Osteoarthritis and Musculoskeleton Research Unit

Faculty of Medicine, Chulalongkorn University,

Bangkok 10330 Thailand.

Tel.  (662) 256-4482 Fax. (662) 256-4482

E-Mail : sittisak.h@chula.ac.th

Round 2

Reviewer 1 Report

I think that issues addressed are discussed in their manuscript. 

Author Response

December 30, 2021

Ms. Fennie Fang,
Assistant Editor, MDPI

Journal: International Journal of Molecular Sciences
Manuscript ID: ijms-1516487
Title: Cytokine Profiling and Intra-articular Injection of Autologous Platelet-Rich Plasma in Knee Osteoarthritis

Dear Editor,

Thank you very much for your mail and the reviewers’ comments on our manuscript entitled “Cytokine Profiling and Intra-articular Injection of Autologous Platelet-Rich Plasma in Knee Osteoarthritis”. We have revised accordingly for all the reviewers’ comments. We also have done following on the check list.

Comments from the editors and reviewers: 

Reviewer #1:

I think that issues addressed are discussed in their manuscript. 

Response: Thank you very much for encouraging advice.

We agree with the entire reviewers’ comments. We hope that all revisions will meet all the reviewers’ recommendation.

Should you have any queries, please do not hesitate to contact me. Along with this letter, we have enclosed the revised manuscript with all changes marked.

Once again, we would like to thank you very much in advance and we are looking forward to hearing from you.

Sincerely yours,

Sittisak Honsawek, MD, PhD

Department of Biochemistry,

Osteoarthritis and Musculoskeleton Research Unit

Faculty of Medicine, Chulalongkorn University,

Bangkok 10330 Thailand.

Tel.  (662) 256-4482 Fax. (662) 256-4482

E-Mail : sittisak.h@chula.ac.th

Reviewer 2 Report

Dear Author,

I think you should add a table in experimental section  containing the sequence  primer that you use in the real time 

best regard

Author Response

December 30, 2021

Ms. Fennie Fang,
Assistant Editor, MDPI

Journal: International Journal of Molecular Sciences
Manuscript ID: ijms-1516487
Title: Cytokine Profiling and Intra-articular Injection of Autologous Platelet-Rich Plasma in Knee Osteoarthritis

Dear Editor,

Thank you very much for your mail and the reviewers’ comments on our manuscript entitled “Cytokine Profiling and Intra-articular Injection of Autologous Platelet-Rich Plasma in Knee Osteoarthritis”. We have revised accordingly for all the reviewers’ comments. We also have done following on the check list.

Comments from the editors and reviewers: 

Reviewer#2

Dear Author,

I think you should add a table in experimental section containing the sequence  primer that you use in the real time 

best regard

Response: Thank you very much for useful advice. We have stated that point in Materials and Methods and “Supplementary Table S2. Primer sequences used for the quantitative real-time PCR analysis”.

We agree with the entire reviewers’ comments. We hope that all revisions will meet all the reviewers’ recommendation.

Should you have any queries, please do not hesitate to contact me. Along with this letter, we have enclosed the revised manuscript with all changes marked.

Once again, we would like to thank you very much in advance and we are looking forward to hearing from you.

Sincerely yours,

Sittisak Honsawek, MD, PhD

Department of Biochemistry,

Osteoarthritis and Musculoskeleton Research Unit

Faculty of Medicine, Chulalongkorn University,

Bangkok 10330 Thailand.

Tel.  (662) 256-4482 Fax. (662) 256-4482

E-Mail : sittisak.h@chula.ac.th

Reviewer 3 Report

While all of my minor comments have been answered adequately, the true major issue was not: there is no connection between the in vitro and the in vivo experiments and as such this paper is merely two pieces of data presented simultaneously. A more relevant in vitro method like what was suggested earlier by this and another reviewer is needed in order to support the conclusions. 

Author Response

December 30, 2021

Ms. Fennie Fang,
Assistant Editor, MDPI

Journal: International Journal of Molecular Sciences
Manuscript ID: ijms-1516487
Title: Cytokine Profiling and Intra-articular Injection of Autologous Platelet-Rich Plasma in Knee Osteoarthritis

Dear Editor,

Thank you very much for your mail and the reviewers’ comments on our manuscript entitled “Cytokine Profiling and Intra-articular Injection of Autologous Platelet-Rich Plasma in Knee Osteoarthritis”. We have revised accordingly for all the reviewers’ comments. We also have done following on the check list.

Comments from the editors and reviewers: 

Reviewer#3

While all of my minor comments have been answered adequately, the true major issue was not: there is no connection between the in vitro and the in vivo experiments and as such this paper is merely two pieces of data presented simultaneously. A more relevant in vitro method like what was suggested earlier by this and another reviewer is needed in order to support the conclusions. 

Response: Thank you so much for very helpful advice. We agree with the valuable comments and suggestions of reviewer. Unfortunately, we did not have enough budgets to perform those methods and did not include those additional experiments into our proposal when submitted to our institutional review board. However, we revised our discussion/ conclusion and addressed those points in Discussion, page 8, line 205-206:

“Therefore, we postulate that the PRP could influence the entire joint environment after intra-articular injection in knee OA patients regardless the effect of cells migration.”

-page 9, line 248-255:

“Previous study outlined that PRP induced diverse effects on articular chondrocytes in vitro, plausibly due to differences in the levels of platelets, leukocytes, growth factors, and other bioactive mediators [42]. We hypothesize that stimulation by the intra-articular injection of PRP could promote the production of cytokines in mesenchymal cells and inflammatory cells of cartilage tissues, possibly leading to the good clinical outcomes. Further studies will be needed to examine whether chondrocytes stimulated by PRP can express these factors or not and to determine the influence of PRP in in vivo animal model experiments.”

-page 9, line 265-269:

“We presume that PRP might influence an anabolic microenvironment, containing the suitable bioactive molecules, which contribute to maintaining the joint homeostasis, reducing pain and improving the articular function. Future in vitro and in vivo experiments are required to investigate the effects of PRP on therapeutic events in OA chondrocytes.”

-And in Conclusion, page 9, line 279-291:

“In conclusion, this study investigated the comprehensive profiling of cytokines, chemokines, as well as growth factors in both PRP and PPP of knee OA subjects and determined its impact on OA chondrocytes in vitro and knee OA patients. Our findings indicated that the concentrations of several growth factors, cytokines, and chemokines in PRP were significantly higher than those in PPP. The addition of PRP further stimulated cell proliferation of OA chondrocytes with potentiated SOX9 transcription resulting in sequentially elevated COL2A1 and aggrecan expression. In knee OA patients, the intra-articular PRP injection significantly reduced pain and improved physical function. We speculate that bioactive molecules in PRP might be an essential component in promoting tissue healing leading to favorable outcomes of knee OA patients. To ensure real-world applicability of PRP, the entire mechanisms underlying the beneficial effects of PRP against knee OA progression should be investigated, in addition to a standardized protocol for PRP preparation.”

-And in References, page 14:

“42. Cavallo C, Filardo G, Mariani E, Kon E, Marcacci M, Pereira Ruiz MT, et al. Comparison of platelet-rich plasma formulations for cartilage healing: an in vitro study. J Bone Joint Surg Am. 2014;96(5):423-9. ”

We agree with the entire reviewers’ comments. We hope that all revisions will meet all the reviewers’ recommendation.

Should you have any queries, please do not hesitate to contact me. Along with this letter, we have enclosed the revised manuscript with all changes marked.

Once again, we would like to thank you very much in advance and we are looking forward to hearing from you.

Sincerely yours,

Sittisak Honsawek, MD, PhD

Department of Biochemistry,

Osteoarthritis and Musculoskeleton Research Unit

Faculty of Medicine, Chulalongkorn University,

Bangkok 10330 Thailand.

Tel.  (662) 256-4482 Fax. (662) 256-4482

E-Mail : sittisak.h@chula.ac.th

Round 3

Reviewer 3 Report

The revision does not contain any further  experimental evidence that would connect the two distinct parts of the manuscript, ie. the in vivo and the in vitro. Without that the conclusion are not adequately supported.